# Preliminary Study on the Purity Analysis of Primary Certified Gas Mixtures Using Different Spectroscopic Techniques

**DOI:** 10.3390/s25196068

**Published:** 2025-10-02

**Authors:** Francesca Rolle, Francesca Durbiano, Stefano Pavarelli, Ramona Russo, Chiara Festevole, Pier Giorgio Spazzini, Francesca Romana Pennecchi, Michela Sega

**Affiliations:** Istituto Nazionale di Ricerca Metrologica, Strada delle Cacce 91, 10135 Torino, Italy; f.durbiano@inrim.it (F.D.); s.pavarelli@inrim.it (S.P.); r.russo@inrim.it (R.R.); chiara.festevole@edu.unito.it (C.F.); p.spazzini@inrim.it (P.G.S.); f.pennecchi@inrim.it (F.R.P.)

**Keywords:** purity data, carbon dioxide, water, metrological traceability, gas sensing

## Abstract

**Highlights:**

**What are the main findings?**

**What is the implication of the main finding?**

**Abstract:**

Purity analysis of parent gases used to produce reference gas mixtures is fundamental to assure the metrological traceability of the certified gas composition, and the use of purity data in the calculation of the mixture composition should be performed in accordance with the requirements of international standards. Purity analysis can be difficult to realize since limited measurement standards are available for the determination of trace levels of gaseous compounds. The first step of purity analysis is the definition of the impurities considered critical or significant to the final composition of a mixture. In this work, we present the activity carried out for the identification and quantification of impurities of carbon dioxide and water in some ultrapure gases used for the preparation of primary certified reference gas mixtures of carbon dioxide at atmospheric amount fraction (400–800 µmol·mol^−1^), by means of different spectroscopic techniques (Fourier Transform IR, Non-Dispersive IR and Cavity Ring-Down). Dynamic dilution was used for the generation of reference mixtures for the calibration of the analyzers by using calibrated Mass Flow Controllers. The certified reference gas mixtures produced with the tested pure gases will also be applied to characterization studies and calibration protocols for gas sensors used both for outdoor and indoor monitoring.

## 1. Introduction

The purity analysis of pure gases is an essential feature in the production of reference gas mixtures to establish the metrological traceability of the certified gas composition, and the use of purity data in the calculation of this composition should be performed in accordance with the requirements of international guidelines, such as the International Standard ISO 19229 [1]. In addition, the determination of impurities contained in the gases used for mixture preparation has an impact on the uncertainty associated with the content of the component [2]. Determination of the purity of pure gases is also required in many scientific sectors, including environmental monitoring, industrial process control, and biomedical diagnosis. Purification of hydrogen and natural gas from carbon dioxide and other contaminants is a fundamental step for the production of green fuels in a climate-neutral economy [3].

Several spectroscopic techniques are applied for this purpose. Fourier transform IR (FTIR), non-dispersive IR (NDIR), and cavity ring-down (CRDS) are widespread spectroscopic techniques useful for purity analysis in gases [4,5,6,7,8,9,10,11]. Other techniques applied to gas sensing include photoacoustic spectroscopy, light-induced thermoelectric spectroscopy, and Raman spectroscopy [12,13,14,15,16,17].

In the field of gas metrology, many national metrology institutes realize primary reference gas mixtures or gaseous certified reference materials by using the gravimetric method as a preparation technique [18,19,20]. The principle of this technique is described in the International Standard ISO 6142-1 [21] and consists of preparation by transferring parent gases (pure gases or gravimetrically prepared mixtures of known composition) quantitatively into appropriately conditioned and treated cylinders in which the gas mixture will be contained. The amount of gas components added in each preparation step is then determined by means of precision weighing to determine the final composition of the mixture from those weighed masses.

Gas composition is preferentially expressed as a mole fraction or amount of substance fraction (mol·mol^−1^). The weighted mass value of each component is used to calculate the number of moles introduced in the cylinder. The amount fraction of each component in the final gas mixture is then given by the quotient of the number of moles of that component to the total number of moles of all of the components present in the gas mixture.

A relevant parameter affecting the accuracy of the composition of the final gas mixture is the purity of the parent mixtures and balance gases used in the gravimetric method. Purity determination can be carried out through the identification and quantification of the impurities in the balance and pure gases [22]. The uncertainty contributions of the balance and pure gases to the final uncertainty of a gas mixture depend on the amount of impurities present in these gases and upon the accuracy with which these impurities have been determined. The analysis and, possibly, quantification of the main impurities in pure gases is a fundamental step in the realization process of a primary gas mixture. [23].

Purity analysis can be difficult to carry out, since limited measurement standards are often available for the determination of trace levels of gaseous compounds. The first step of a purity analysis procedure is to identify which impurities should be considered critical, or significant, to the final composition of a mixture [1]. Particular attention should be paid to the identification of the significant impurities in the pure gases and the comparison of the results obtained with the technical specifications of the gas producers.

In addition, the use of dynamic methods (e.g., permeation and diffusion) can be a valid approach to generate reference gas mixtures at trace amount fractions. Another possibility is represented by dynamic dilution, where a reference gas mixture at a higher amount fraction than the generated mixture, which is stable in high-pressure cylinders, is continuously introduced into a diluent gas in a controlled flowing system. This approach can significantly reduce losses of the target component from adsorption onto internal surfaces, with respect to a static system [24].

In this work, we present some activities carried out for the preliminary identification and quantification of impurities of carbon dioxide (CO_2_) and water in ultrapure gases used for the preparation of primary Certified Reference Materials (CRMs) of CO_2_ in synthetic air (SA) at atmospheric amount fraction (400–800 µmol·mol^−1^), by means of FTIR, NDIR and CRDS spectroscopies. These techniques were chosen as they were previously successfully applied to the quantification of the CO_2_ amount fraction in the certified reference gas mixtures prepared by means of gravimetry. CO_2_ is considered a significant impurity, being the main analyte of the CRMs prepared, and its amount fraction in the matrix gases is accounted for in the calculations of the composition of the final gas mixtures. H_2_O impurities should also be monitored to avoid possible undesired side reactions inside the cylinders, which could cause the earlier degradation of the mixture compounds and compromise the stability of the CRMs over time.

It was decided to start the study from ultrapure N_2_ and argon (Ar) (grade 6.0 and 5.0, respectively), as these gases are among the main components of the air matrix of the gravimetric gas mixtures and represent the typical pure gases used in our laboratory in the preparation of certified reference gas mixtures by means of gravimetry. In addition, the purity grade of the gases under study was chosen as it is the best compromise between an extremely high level of purity and the high commercial cost.

Dynamic dilution was used for the generation of ad hoc reference mixtures for the calibration of the analyzers by using a system made up of calibrated Mass Flow Controllers (MFCs) and a mixing chamber.

The scope of this work is the application of well-established spectroscopic techniques to the analysis of the purity of parent pure gases, used in the preparation of primary certified reference gas mixtures (CRMs) by means of gravimetry. The preliminary results obtained in our laboratory confirm the suitability of the methods for assessing the quality, accuracy, and long-term stability of the gaseous CRMs. This point is also of utmost importance for participation in International Key Comparisons, to support Calibration and Measurement Capabilities. The results are reported in this paper together with a preliminary uncertainty evaluation.

Although the topic addressed in the manuscript is not directly related to sensor development, the theme of purity analysis of pure gases for certified reference mixture production is also of great relevance in the sensors field and is propaedeutic to their development and metrological characterization.

The proposed metrological approaches represent a solid basis both for the qualification of the CRMs of CO_2_ at atmospheric amount fraction and for the characterization of gas sensors for atmospheric monitoring.

## 2. Materials and Methods

### 2.1. CO_2_ Impurity Determination via FTIR

This approach, based on FTIR spectroscopy, was applied to the evaluation of CO_2_ impurities in ultra-pure N_2_ grade 6.0 (99.9999%) from Messer, Italy, used for gas mixture preparation by means of gravimetry. The instrument used was a Thermo Fisher Scientific Nicolet iS50, equipped with a gas cell with a path length equal to 2 m (shown in Figure 1). The FTIR spectra were acquired with a resolution of 1 cm^−1^, and each spectrum was acquired with a scan time of 224 s. Additional information on FTIR parameters and an example of spectrum can be found as Appendix A.

The gravimetric method allows to prepare mixtures in high-pressure cylinders by weighing and subsequently mixing the different components of the gas mixture [9].

For the evaluation and quantification of the CO_2_ impurities in N_2_, the application of dynamic dilution was exploited to reach very low amount fractions of CO_2_ in N_2_, used as diluent gas. The dilution approach to estimate the impurities in the ultrapure N_2_ is based on the standard addition concept [25], by spiking known CO_2_ amount fractions in the pure gas under test. This method has several advantages, in particular the reduction of matrix effects and the possibility to carry out measurements away from the detection limit. Two MFCs, with full-scale ranges of 200 sccm (standard cubic centimeters per minute) and 2000 sccm, respectively (MKS Instruments, USA), were employed to generate reference mixtures in the range (3.5–13) μmol·mol^−1^, starting from a parent mixture of CO_2_ in N_2_ at the amount fraction of 499.56 μmol·mol^−1^ and an expanded uncertainty of 0.4 μmol·mol^−1^ (*k* = 2). The total flow used for the generation of the reference mixtures was 1200 sccm (corresponding to ~80 L·h^−1^). The selection of the two flow rates for the dynamic dilution was carried out by means of a previous optimization study to obtain a total flow rate that was close to the one required by the NDIR analyzer when working in sampling mode. The same flow rate was then tested on the FTIR instrument, showing a good signal stability after 30 min of acquisition and efficient flushing of the gas cell, with minimization of the interferences from the environment in the background. The dilution system with this configuration allows for the generation of reference mixtures of CO_2_ with a relative standard uncertainty in the range (0.5–0.01) %, obtaining an increasing accuracy with increasing flow rate.

In Table 1, the nominal flows (in sccm) used to obtain the different CO_2_ amount fraction levels, to be analyzed by means of FTIR, are reported with their associated standard uncertainty.

The model equation for dynamic dilution is as follows:***χ***_out_ = (*Q*_1_·***χ***_1_ + *Q*_2_·***χ***_2_)/(*Q*_1_ + *Q*_2_)(1),
where:

*Q*_1_ is the flow generated by the MFC with a full-scale range of 200 sccm

***χ***_1_ is the amount fraction of the parent gas mixture of CO_2_ at a higher amount fraction

*Q*_2_ is the flow generated by the MFC with a full-scale range of 2000 sccm

***χ***_2_ is the amount fraction of CO_2_ (as impurity) in the pure N_2_.

### 2.2. CO_2_ Impurity Determination via NDIR

NDIR spectroscopy can be useful in detecting low levels of gases absorbing IR radiation and has the advantage of being a robust and selective technique for the analyte chosen. For this work, a NDIR ABB URAS 14 was used to detect small amounts of CO_2_ in pure argon (Ar, 5.0 grade) that was used in the preparation of CO_2_ in synthetic air mixtures, at atmospheric amount fraction (400–800 µmol·mol^−1^). Additional information on NDIR parameters can be found as Appendix A. The dynamic dilution technique was used to generate the mixtures for the calibration of the NDIR analyser, and the details are reported in Table 2. The NDIR instrument was calibrated in the amount fraction range (0.199–0.714) µmol mol^−1^ by diluting a parent mixture of CO_2_ in synthetic air at 49.97 µmol·mol^−1^. The set of MFCs used in this case was 500 sccm for the parent mixture and 2000 sccm for the diluent gas (synthetic air). The total flow used was 1200 sccm. The nominal CO_2_ amount fractions considered for calibration are reported in Table 2, with the corresponding standard uncertainties, and the flows used for dynamic dilution. Each calibration mixture was analyzed 5 times, for 3 consecutive days, and each measurement was acquired for 90 s. The NDIR instrument showed good linearity in the range considered, as reported in Section 3.

The pure Ar under test was then analyzed with the calibrated NDIR instrument, by using the 2000 sccm MFC to regulate the gas flow, by imposing a total flow of 863 sccm, considering the gas conversion factor for pure Ar (equal to 1.39). In Figure 2, the system used for the CO_2_ impurity analysis is shown. The NDIR analyzer is equipped with two gas lines, A and B, from which the pure N_2_ for zero measurement and the different calibration mixtures, generated by means of dynamic dilution, are supplied.

### 2.3. H_2_O Impurity Determination via CRDS

A CRDS analyzer (G2131-i, Picarro, USA) devoted to the analysis of CO_2_ stable isotopes and also able to detect H_2_O in air samples, was applied to the detection of H_2_O traces in gaseous CRMs of CO_2_ in air at atmospheric amount fraction.

CRDS is a highly sensitive laser-based absorption technique with advantages such as high accuracy and suitability for trace gas analysis [26,27]. CRDS uses a laser in a sample cell (cavity) containing the gas sample. The laser light bounces back and forth between highly reflective mirrors in the cavity, generating a long effective path length. When a molecule (as CO_2_ or H_2_O) absorbs light at a specific wavelength, the light intensity in the cavity decays more rapidly. By measuring this decay time (or “ring-down” time), CRDS can determine the concentration of the absorbing molecule.

The analyses were conducted on several mixtures, with a nominal amount fraction of around 400 µmol·mol^−1^, which were prepared by means of gravimetry in different types of cylinders with and without internal coating, to improve the mixtures’ stability. The acquisition time for each mixture was 10 min to allow for the stabilization of the readings. The instrument acquires 1 scan/second, and the last 100 acquisitions were considered for the calculation of the average value of each mixture. The measurement sequence was repeated 4 times to also consider instrumental repeatability. Additional information on CRDS parameters can be found as Appendix A.

This analysis was qualitative, as it was not possible to calibrate the instrument for the H_2_O content, but it allowed us to obtain useful information about the dryness of the prepared gas mixtures and of the entire measurement system used, depicted in Figure 3. This is important information on the quality of the prepared CRMs, however, to avoid possible side-products in the mixtures and reduced stability of their composition over time.

In Table 3, we report the sequence of cylinders that were analyzed for the CO_2_ and H_2_O amount fractions by means of CRDS, together with the type of air matrix for each cylinder.

## 3. Results

### 3.1. CO_2_ Impurities via FTIR

In Figure 4, the good linearity of the FTIR device in the investigated range is shown. As previously mentioned, the FTIR device was calibrated at low amount fraction levels in the range (3.5–13) μmol·mol^−1^. A linear calibration was carried out by means of the Weighted Total Least Squares (WTLS) algorithm, following the model Equation (2):*y* = *a* + *b·**χ_s_***_td_(2)
where:

*y* is the response signal of the FTIR, expressed in amount fraction (μmol·mol^−1^)

*a* is the intercept of the calibration straight line (μmol·mol^−1^)

*b* is the slope of the calibration straight line (adimensional)

***χ_s_***_td_ is the amount fraction of the calibration mixtures (μmol·mol^−1^)

To obtain the amount fraction values of unknown mixtures, it is necessary to reverse Equation (2), obtaining the analysis curve in Equation (3):***χ_s_***_td_ = (*y* − *a*)/*b*(3)

In this case, to extrapolate the value of CO_2_ impurities in the N_2_ 6.0 from the analysis curve ***χ***_CO2_, we consider the following model equation, as prescribed in the standard addition method [13]:***χ***_CO2_ = *a*/*b*,(4)
where:

*a* is the intercept of the calibration straight line

*b* is the slope of the calibration straight line

***χ***_CO2_ is the extrapolated amount fraction for the CO_2_ impurities (corresponding to *y* = 0)

By applying the WTLS regression, the values and uncertainties of the coefficients *a* and *b* were evaluated, together with the covariance among the two parameters. Those values, reported in Table 4, were used for the subsequent evaluation of the measurement uncertainty of *χ*_CO2_ in pure N_2_ by the LPU and for the MCM simulation.

From Equation (4), ***χ***_CO2_ in pure N_2_ was calculated as 0.06 μmol·mol^−1^. This value is compliant with the manufacturer specifications for the pure N_2_ 6.0, equal to CO_2_ ≤ 0.1 μmol·mol^−1^. In Figure 4, the calibration curve obtained for the CO_2_ in the N_2_ amount fraction, in the range (3.5–13) μmol·mol^−1^, is shown. The error bars correspond to the standard deviation of *n* = 6 repeated measurements of each calibration reference mixture.

**Figure 4 sensors-25-06068-f004:**
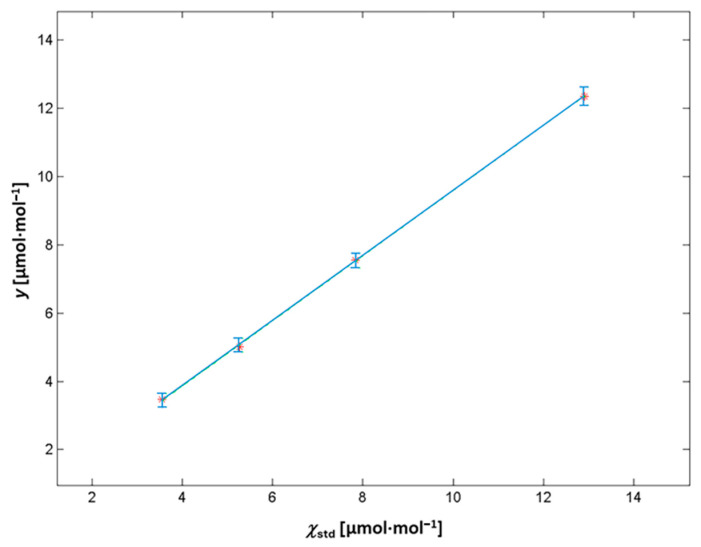
Calibration curve for CO_2_ in N_2_ impurities using FTIR.

The expanded uncertainty associated with ***χ***_CO2_ was evaluated following the approach based on the Law of Propagation of Uncertainties (LPU) [28], and the propagation of distributions by means of the Monte Carlo method (MCM) [29].

Considering the model equation in Equation (4), we obtained an expanded uncertainty *U*(*χ*_CO2_) with the LPU approach equal to 0.32 μmol·mol^−1^ (*k* = 2), dominated by the dilution process and the extrapolation at very low amount fraction levels. Note that the corresponding coverage interval for the measurand would then be [−0.26, 0.38] μmol·mol^−1^, hence encompassing unfeasible negative concentration values. This might happen when relative uncertainty is large (i.e., measurand value close to 0). By applying the MCM (with a truncation at 0 of simulated values, thus neglecting possible negative values obtained for *χ*_CO2_ in the MC simulation) [30], a whole probability density function (PDF) for ***χ***_CO2_ is obtained, as shown in Figure 5. Because of the evident asymmetry of the distribution, it is not meaningful to construct a symmetric interval [*χ*_CO2_ − *U*, *χ*_CO2_ + *U*] where the expanded uncertainty is obtained by multiplying the MCM standard uncertainty (*u* = 0.11 μmol·mol^−1^) by some coverage factor. The guidelines in [29] recommend calculating a coverage interval, directly from the obtained PDF, such that it encompasses a desired fraction of the simulated values for the measurand. The black bars in Figure 5 indicate the range of the most plausible values for the measurand ***χ***_CO2_, at a coverage level of 95%, that is, the interval [0, 0.35] μmol·mol^−1^ is the shortest 95% coverage for the measurand. Also in this case, the coverage interval is influenced by the extrapolation and the dilution process.

### 3.2. CO_2_ Impurities via NDIR

The NDIR was calibrated in the range (0.199–0.714) µmol mol^−1^, as reported in Section 2. The methodological approach was analogous to the one used for the FTIR calibration, but with no need to extrapolate the final impurity values in the pure Ar tested. In Figure 6, the calibration straight line obtained is shown, and the error bars represent the standard deviations of the repeated measurements for each calibration mixture (*n* = 15).

By applying WTLS regression, the values and uncertainties of the coefficients *a* and *b* were evaluated, together with the covariance among the two parameters. Those values, reported in Table 5, were used for the subsequent evaluation of the measurement uncertainty of *χ*_CO2_ in pure Ar by the LPU.

The CO_2_ impurities in the pure Ar were evaluated to be 0.36 µmol mol^−1^, and this value is compliant with the manufacturer’s specifications for pure Ar grade 5.0 (99.999%), ≤0.5 μmol·mol^−1^. The associated expanded uncertainty (*k* = 2) has a value of 0.13 μmol·mol^−1^, which also dominated in this case by the dilution process.

### 3.3. H_2_O Impurities via CRDS

The analysis of a set of CRMs of CO_2_ in air matrix at atmospheric amount fraction, to detect the impurities of H_2_O in the cylinders by means of CRDS, resulted in very low levels of H_2_O, comprising the range (0.034–0.046) µmol·mol^−1^, which are desirable for the quality and stability of the CRMs over their lifetime. In Figure 7, the results are reported showing similar levels of H_2_O in mixtures with similar matrices and preparation dates (see Table 3). The uncertainty bars are the standard deviations for 4 repeated measurements, multiplied by a coverage factor *k* = 2.

In Figure 7, the names assigned to the analyzed gas mixtures are reported on the *x*-axis. Those mixtures were realized in the framework of two European projects, namely, the EMPIR projects 16ENV06 “SIRS” and 19ENV05 “STELLAR”, and the names were used to label the gas mixtures.

The agreement between the results among the different sets of cylinders is encouraging for the assurance of the optimal preparation procedure of the CRMs and the stability of the mixtures over time.

## 4. Discussion and Conclusions

The results presented show the possibilities offered by the well-established spectroscopic techniques FTIR, NDIR, and CRDS for the analysis of impurities of CO_2_ and H_2_O at low amount fraction levels, following sound metrological approaches, in the field of gaseous CRMs’ realization. In Figure 8, the uncertainty budget for a CO_2_ in synthetic air mixture at atmospheric amount fraction (413.1 µmol·mol^−1^, *U*(*χ*) = 0.4 µmol·mol^−1^, *k* = 2) is reported, to visually highlight the relevance of the contributions coming from the different pure gases constituting the matrix of a gas mixture to its overall uncertainty.

At the European and international level, few measurement comparisons have been carried out to quantify impurities in matrix gases and to evaluate the comparability of purity analyses performed by participating laboratories [31]. In this framework, the efforts to improve the accuracy and precision of the purity analysis for reference gas mixtures’ preparation and certification are of utmost importance, also by comparing results obtained with different spectroscopic techniques.

In this paper, a preliminary evaluation of the purity of matrix gases used for primary gas mixtures and CRM preparation is presented. N_2_, Ar, and synthetic air were chosen, with them being among the components of the matrices of the prepared gas mixtures.

CO_2_ and H_2_O impurities in the parent gases are important when realizing diluted CO_2_ gas mixtures characterized by both the amount fraction and the isotopic composition.

Novel aspects in the present work are related to the methodology used for data analysis, in particular, the comparison between the LPU approach and the MCM, applied to the uncertainty evaluation of the impurities of CO_2_ (*χ*_CO2_) in the pure N_2_ matrix gas. The application of MCM for uncertainty evaluation in analytical chemistry is not yet commonly implemented, though it constitutes a valid alternative to the LPU approach, in particular in the case of a large relative uncertainty associated with the measurement result. Unreliable expanded uncertainty intervals should be avoided and substituted with feasible coverage intervals, such as those provided by the MCM.

Concerning the characterization results of the gases, the applied methods allowed us to confirm the high quality of the pure gases used, in particular in terms of CO_2_ impurities, and the compliance with the manufacturers’ specifications. This is of paramount importance for the quality and stability of the prepared CRMs. This point is also fundamental for participation in International Key Comparisons, to support Calibration and Measurement Capabilities listed in the BIPM database (https://www.bipm.org/kcdb/cmc/quick-search, accessed on 18 September 2025), as it is not convenient to rely only on the information provided by the manufacturers.

As for H_2_O, a qualitative investigation on the presence of trace H_2_O in a set of primary CO_2_ in SA gas mixtures was conducted, with promising outcomes.

Future investigation of other matrix gases (e.g., oxygen) and impurities (e.g., CH_4_, CO, and N_2_O) is foreseen by exploiting the different available spectroscopic techniques. In addition, the combined use of FTIR, NDIR, and CRDS will also be tested for purity analysis. The use of purity data will improve the accuracy of the amount fraction value assigned to the primary certified reference gas mixtures realized by means of gravimetry. The proposed metrological approaches thus represent a solid basis both for the qualification of the CRMs of CO_2_ at atmospheric amount fraction and for their application in the characterization and calibration of gas sensors for the monitoring of CO_2_ levels in both outdoor and indoor conditions.

## Figures and Tables

**Figure 1 sensors-25-06068-f001:**
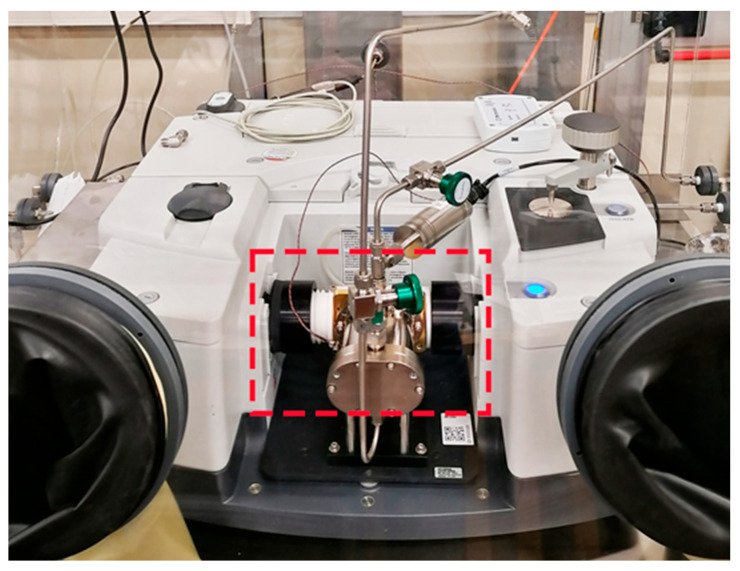
FTIR Thermo Fisher Scientific Nicolet iS50 equipped with a 2 m multi-pass gas cell highlighted in the dotted red box.

**Figure 2 sensors-25-06068-f002:**
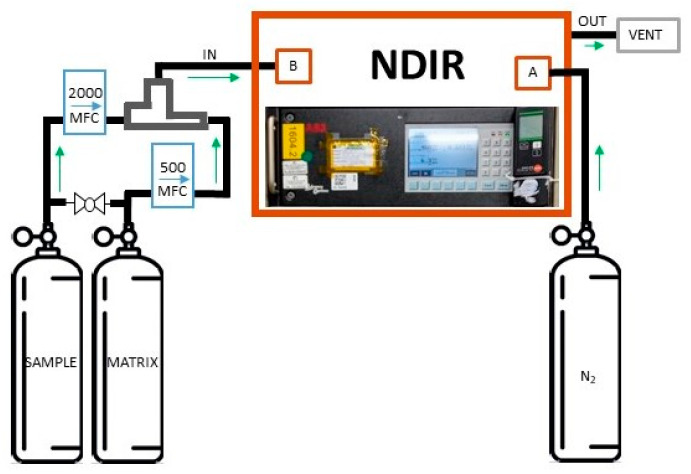
Schematic representation of the set-up of the NDIR analyzer URAS 14 (ABB, Switzerland) for the analysis of CO_2_ amount fractions in pure Ar.

**Figure 3 sensors-25-06068-f003:**
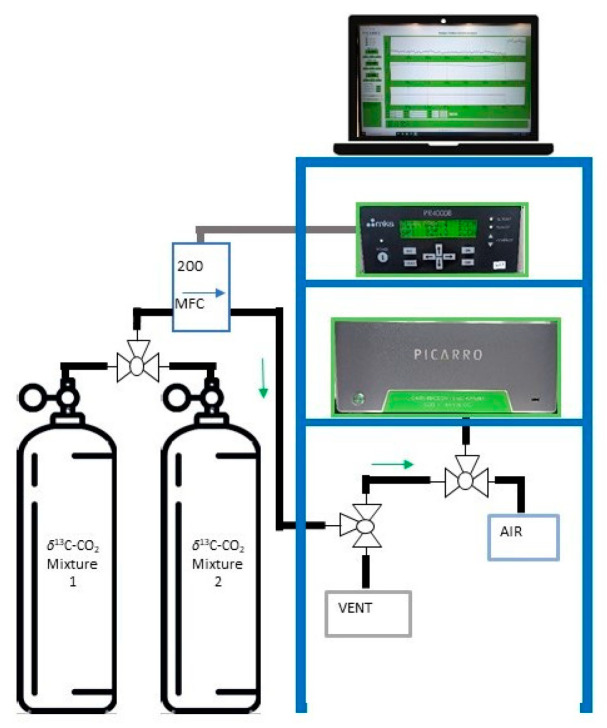
Schematic representation set-up of the CRDS analyzer G2131-i (Picarro, USA) for the analysis of CO_2_ stable isotopes and H_2_O content in air samples.

**Figure 5 sensors-25-06068-f005:**
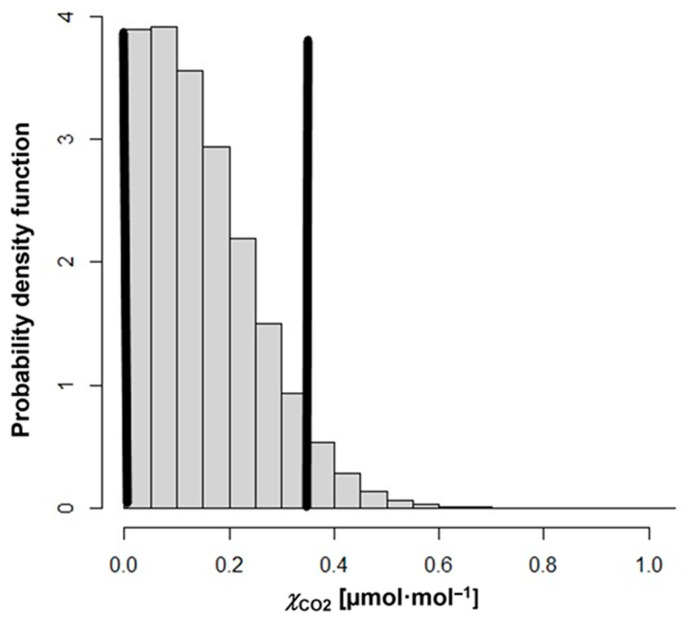
Probability density function (PDF) obtained for *χ*_CO2_ using MCM. The black bars indicate the shortest 95% coverage interval for the measurand *χ*_CO2_.

**Figure 6 sensors-25-06068-f006:**
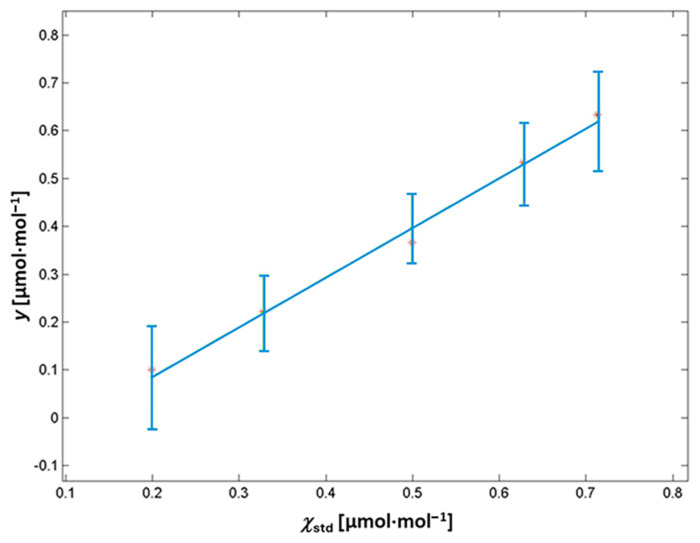
Calibration curve for the CO_2_ in Ar impurities using NDIR, with the error bars representing the standard deviations of the repeated measurements for each calibration mixture (*n* = 15).

**Figure 7 sensors-25-06068-f007:**
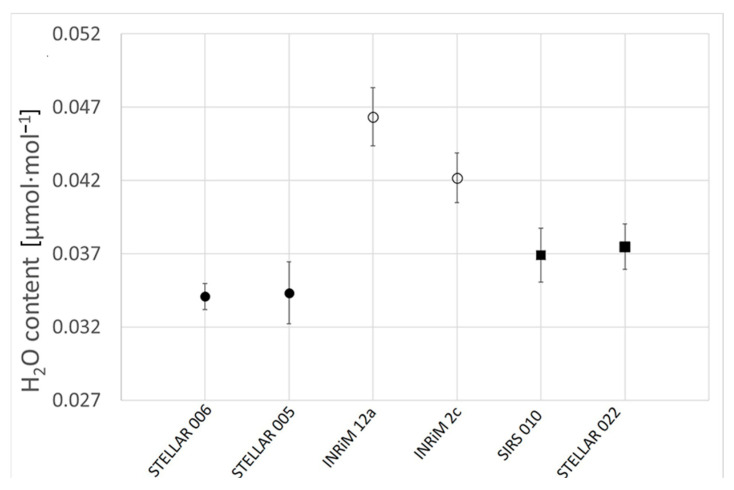
Comparison of the results obtained for the H_2_O content (μmol·mol^−1^) in different CO_2_ in air reference gas mixtures, measured using CRDS. Different symbols are used to denote pairs of mixtures having similar matrices and preparation dates.

**Figure 8 sensors-25-06068-f008:**
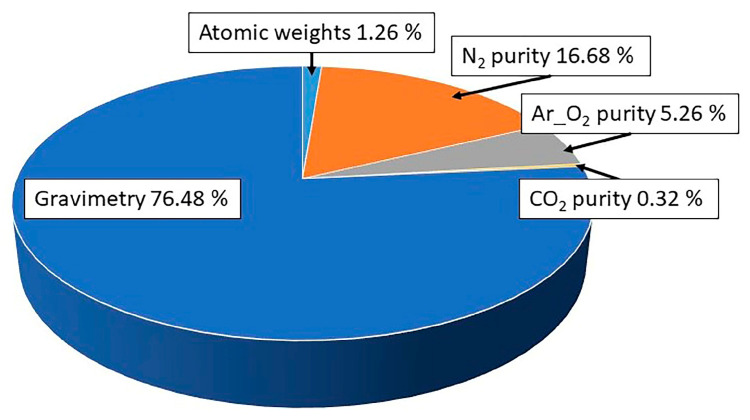
Example of uncertainty components in a certified reference mixture (CO_2_ in SA at 413.1 µmol·mol^−1^), considering the relative contributions (%) from the different pure gases used in the matrix to the final uncertainty of the mixture.

**Table 1 sensors-25-06068-t001:** Nominal flow (in sccm) used to obtain the different CO_2_ amount fraction levels, to be analyzed using FTIR, with their associated standard uncertainty.

Flow *Q*_1_ (CO_2_ in N_2_ Parent Mixture)	Flow *Q*_2_ (Pure N_2_ grade 6.0)	CO_2_ Amount Fraction *χ*_out_	*u*(*χ*_out_)
sccm	sccm	µmol mol^−1^	µmol mol^−1^
8	1192	3.55	0.06
12	1188	5.27	0.06
18	1182	7.83	0.06
30	1170	12.90	0.06

**Table 2 sensors-25-06068-t002:** Nominal flow (in sccm) used to obtain the different CO_2_ amount fraction levels, for the calibration of the NDIR, with their associated standard uncertainty.

Flow *Q*_1_ (CO_2_ in SA Parent Mixture)	Flow *Q*_2_ (Synthetic Air Grade 5.7)	CO_2_ Amount Fraction *χ*_out_	*u* (*χ*_out_)
sccm	sccm	µmol mol^−1^	µmol mol^−1^
7	1193	0.199	0.003
10	1190	0.328	0.005
14	1186	0.499	0.008
17	1183	0.628	0.010
19	1181	0.714	0.011

**Table 3 sensors-25-06068-t003:** List of cylinders analyzed using CRDS, with the corresponding identification code, CO_2_ amount fraction, and air matrix type.

Cylinder Code	Matrix Type	Preparation Date	CRDS CO_2_ Amount Fraction µmol mol^−1^	CRDS H_2_O Amount Fraction µmol mol^−1^
STELLAR 006	Scrubbed natural air	27 January 2022	406.72	0.0341
STELLAR 005	Scrubbed natural air	27 January 2022	406.73	0.0343
INRiM 12A	Synthetic air (with 1% Ar)	9 August 2024	403.56	0.0463
INRiM 2C	Synthetic air (with 1% Ar)	26 July 2024	413.45	0.0422
SIRS 010	Synthetic air (with 1% Ar)	13 January 2020	387.05	0.0369
STELLAR 022	Synthetic air (with 1% Ar)	4 April 2022	402.09	0.0375

**Table 4 sensors-25-06068-t004:** Values and standard uncertainties *u*(*a*) and *u*(*b*) of the coefficients *a* and *b* and their covariance *u*(*a, b*). *a* and *b* are the coefficients obtained from the WTLS regression for the FTIR method.

Quantity	Quantity Value	Standard Uncertainty
Intercept of the calibration curve, *a*	0.06 μmol·mol^−1^	0.15 μmol·mol^−1^
Slope of the calibration curve, *b*	0.954	0.016
		**Covariance *u*(*a*, *b*)**
		−0.0023

**Table 5 sensors-25-06068-t005:** Values and standard uncertainties *u*(*a*) and *u*(*b*) of the coefficients *a* and *b*, and their covariance *u*(*a, b*). *a* and *b* are the coefficients obtained from the WTLS regression for the NDIR method.

Quantity	Quantity Value	Standard Uncertainty
Intercept of the calibration curve, *a*	0.121 μmol·mol^−1^	0.081 μmol·mol^−1^
Slope of the calibration curve, *b*	1.04	0.15
		**Covariance *u*(*a*, *b*)**
		−0.012

## Data Availability

The authors declare that the data supporting the findings of this study are available within the paper. Should any raw data files be needed in another format, they are available from the corresponding author upon request.

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
