# Peer review of "Preliminary Study on the Purity Analysis of Primary Certified Gas Mixtures Using Different Spectroscopic Techniques"

_sensors, 2025, doi:10.3390/s25196068_

Round 1
Reviewer 1 Report
Comments and Suggestions for Authors
This manuscript systematically investigates the analysis of COâ‚‚ and H2O impurities in ultra-pure gases (N2, Ar, and synthetic air) using three techniques: Fourier-transform infrared spectroscopy (FTIR), non-dispersive infrared spectroscopy (NDIR), and cavity ring-down spectroscopy (CRDS). It achieves precise measurement of gas purity and evaluates its impact on the certification of primary certified reference gas mixtures (CRMs). The study is well-designed, the experimental methods are rigorous, the data are comprehensive and reliable, and the analysis and discussion are thorough. The authors innovatively combine dynamic dilution techniques with spectroscopic analysis methods, successfully achieving quantitative detection of trace impurities in ultra-pure gases and qualitative analysis. The research findings are of great significance for improving the accuracy and reliability of certified reference gas mixtures. The paper is well-written, the figures and tables are clear, and the conclusions are robust. It is recommended for publication after minor revisions to supplement some details:
Q1: The authors should clearly specify the concentration levels of the measured CO2 gas used in the experiments. Furthermore, it should be explicitly stated whether the measured CO2 was in its pure form or prepared at specific volume fractions. This information is crucial for understanding the experimental conditions and ensuring reproducibility of the results.
Q2: The experimental setup utilizes a relatively low flow rate (200 sccm) for the CO2 sample gas while employing a significantly higher flow rate (2000 sccm) for the background gas (N2/Ar). This configuration raises several technical questions that require clarification: What was the rationale behind selecting these specific flow rate ratios? Please explain how this configuration optimizes the measurement sensitivity and accuracy for COâ‚‚ detection.
Q3: The article mentions that the ultra-pure gases studied were N2 (grade 6.0) and Ar (grade 5.0). What was the specific basis for selecting these two grades of gas as the subjects of the study?
Q4: When analyzing H2O impurities using CRDS, only qualitative analysis was performed, and the instrument was not calibrated. Could this affect the reliability of the Hâ‚‚O impurity content data?
Q5: There are many kinds of spectroscopy methods can be used for gas detection. Why these three method of FTIR, NDIR, and CRDS are used? Furthermore, other common used techniques of photoacoustic spectroscopy, light-induced thermoelastic spectroscopy, and Raman spectroscopy should be added in the manuscript to give readers a more complete introduction. [Adv. Photonics. 2024, 6, 066008], [ Light Sci. Appl. 2025, 14, 180], [Opto-Electron. Adv. 2023, 6, 230094].
Author Response
REVIEWER 1
Comments and Suggestions for Authors
This manuscript systematically investigates the analysis of COâ‚‚ and H2O impurities in ultra-pure gases (N2, Ar, and synthetic air) using three techniques: Fourier-transform infrared spectroscopy (FTIR), non-dispersive infrared spectroscopy (NDIR), and cavity ring-down spectroscopy (CRDS). It achieves precise measurement of gas purity and evaluates its impact on the certification of primary certified reference gas mixtures (CRMs). The study is well-designed, the experimental methods are rigorous, the data are comprehensive and reliable, and the analysis and discussion are thorough. The authors innovatively combine dynamic dilution techniques with spectroscopic analysis methods, successfully achieving quantitative detection of trace impurities in ultra-pure gases and qualitative analysis. The research findings are of great significance for improving the accuracy and reliability of certified reference gas mixtures. The paper is well-written, the figures and tables are clear, and the conclusions are robust. It is recommended for publication after minor revisions to supplement some details:
The authors would like to thank the reviewer for the comments on the manuscript. In the following, a point-by-point response to the queries is reported.
Q1: The authors should clearly specify the concentration levels of the measured CO2 gas used in the experiments. Furthermore, it should be explicitly stated whether the measured CO2 was in its pure form or prepared at specific volume fractions. This information is crucial for understanding the experimental conditions and ensuring reproducibility of the results.
Different types of CO2 samples were used for the experiments with the different analytical techniques. In the case of N2, for the calibration of the FTIR, using the standard additions method, a CO2 in N2 parent mixture at an amount fraction value of 499.56 µmol mol-1, was used for the dynamic dilution. In the case of the analysis of CO2 impurities in Ar by NDIR, the instrument was calibrated with CO2 in N2 mixtures obtained by dynamic dilution from a parent mixture of CO2 in N2 at 49.97 µmol·mol-1, while the sample analysed were of pure Ar. Finally, for the analyses with CRDS, CO2 in synthetic air mixtures at a nominal amount fraction of around 400 µmol·mol-1 were analysed to check for trace H2O impurities. The missing details were added in the “Materials and methods” section.
Q2: The experimental setup utilizes a relatively low flow rate (200 sccm) for the CO2 sample gas while employing a significantly higher flow rate (2000 sccm) for the background gas (N2/Ar). This configuration raises several technical questions that require clarification: What was the rationale behind selecting these specific flow rate ratios? Please explain how this configuration optimizes the measurement sensitivity and accuracy for COâ‚‚ detection.
The selection of the two flow rates for the dynamic dilution was carried out by means of a previous optimization study, to obtain a total flow rate that was close to the one required by the NDIR analyser, when working in sampling mode. The flow rate required by the NDIR analyser was equal to 1200 sccm (corresponding to 80 L/h). The same flow rate was then tested on the FTIR, showing a good signal stability after 30 min of acquisition, and an efficient flushing of the gas cell, with minimisation of the interferences from the environment in the background. The dilution system with this configuration allows to generate reference mixtures of CO2 having a relative standard uncertainty in the range (0.5–0.01) %, obtaining an increasing accuracy with increasing flow rate. This point was clarified in section 2.1.
Q3: The article mentions that the ultra-pure gases studied were N2 (grade 6.0) and Ar (grade 5.0). What was the specific basis for selecting these two grades of gas as the subjects of the study?
The purity grade of the gases under study, N2 (grade 6.0) and Ar (grade 5.0), was chosen as these are the typical pure gases used in our laboratory in the preparation of certified reference gas mixtures by gravimetry. The purity grade of the gases was chosen as it is the best compromise between an extremely high level of purity and the high commercial cost. This point was specified in the Introduction of the paper.
Q4: When analyzing H2O impurities using CRDS, only qualitative analysis was performed, and the instrument was not calibrated. Could this affect the reliability of the Hâ‚‚O impurity content data?
We agree that the analyses carried out on the H2O content of the gas mixtures can give only indications on the overall dryness of the pure gases and of the final mixture. For the purposes of the authors, this gives important information on the quality of the prepared CRMs, to avoid possible side-products in the mixtures and reduced stability of their composition overtime. This point was also highlighted in the text, in section 2.3.
Q5: There are many kinds of spectroscopy methods can be used for gas detection. Why these three method of FTIR, NDIR, and CRDS are used? Furthermore, other common used techniques of photoacoustic spectroscopy, light-induced thermoelastic spectroscopy, and Raman spectroscopy should be added in the manuscript to give readers a more complete introduction. [Adv. Photonics. 2024, 6, 066008], [ Light Sci. Appl. 2025, 14, 180], [Opto-Electron. Adv. 2023, 6, 230094].
FTIR, NDIR, CRDS techniques were chosen as they were previously successfully applied to the quantification of CO2 amount fraction of the certified reference gas mixtures prepared by gravimetry.
We acknowledge that several other techniques are commonly used for this purpose, and we added some of the suggested references in the text.
Reviewer 2 Report
Comments and Suggestions for Authors
Gas sensor specialists use reference gas mixtures, so it is interesting for me and my colleagues to learn about the methods used to prepare them. This manuscript focuses on analyzing the purity of gases in reference gas mixtures.
I did not find any errors or inaccuracies in the manuscript, and I did not have any questions for the authors regarding their research. However, it should be noted that the novelty of these studies is limited, as the authors do not develop new methods or create new devices, but rather use standard methods.
In addition, it should be noted that the journal is called Sensors, and the authors in their study not only did not develop new sensors, but even did not use standard sensors in their research. The reviewer did not find any factual errors in the manuscript, but the editors must decide whether these studies are sufficiently novel and whether they are in line with the scope of the Sensors journal.
Author Response
REVIEWER 2
Comments and Suggestions for Authors
Gas sensor specialists use reference gas mixtures, so it is interesting for me and my colleagues to learn about the methods used to prepare them. This manuscript focuses on analyzing the purity of gases in reference gas mixtures.
I did not find any errors or inaccuracies in the manuscript, and I did not have any questions for the authors regarding their research. However, it should be noted that the novelty of these studies is limited, as the authors do not develop new methods or create new devices, but rather use standard methods.
In addition, it should be noted that the journal is called Sensors, and the authors in their study not only did not develop new sensors, but even did not use standard sensors in their research. The reviewer did not find any factual errors in the manuscript, but the editors must decide whether these studies are sufficiently novel and whether they are in line with the scope of the Sensors journal.
The authors would like to thank the reviewer for the comments on the paper. We acknowledge that the topic addressed in the manuscript is not directly related to sensors development, but we think that the theme of purity analysis of pure gases for certified reference mixture production is of great relevance also in the sensors field and is propaedeutic to their development and metrological characterisation. We chose to submit the paper to the special issue entitled “Advanced Sensors for Gas Monitoring: 2nd Edition” due to the broad scope of this SI, and in particular as the core themes of the paper fit well in the point “gas measurements for environmental applications”. In addition, the proposed metrological approaches could represent a solid basis for the characterization of gas sensors for atmospheric monitoring.
Reviewer 3 Report
Comments and Suggestions for Authors
The research topic may be relevant, but overall, the manuscript isn't well readable. Some substantive data is missing, as specified below. The novelty and motivation are almost absent. There are also some repetitive parts, which can be phrased more concisely.
- I'm a bit confused, which gases are to be considered as the main gases (i.e., intentional constituents of a prepared gas mixture) and which ones are to be considered as impurities (and why). Perhaps these two aspects are even used interchangably in the manuscript.
- I may be mistaken, but high-purity gases don't seem relevant for carbon capture and storage research (second paragraph of introduction). Maybe it would be relevant to mention some other typical applications of these traceable gas mixtures from the outset.
- The novelty of the work remains vague. CO2 and water vapor are rather common substances for spectroscopic investigation. Are there any improvements of accuracy or sensitivity compared to other studies?
- Regarding the terminology, "amount fraction" sounds a bit clumsy to me. Isn't it the same as "molar fraction"? The term "ambient amount fraction" requires clarification as well.
- Similarly, instead of "µmol mol-1" I would prefer "ppm" (parts per million), but the usage may depend on the research area.
- I suppose NDIR and CRDS techniques involve specific wavelengths/spectral bands, where they operate (depending on the target molecule). I didn't find any information on that. Moreover, FTIR presumably involves spectral measurements, but I don't see any infrared spectra or related information. Some data could be included as supplementary material.
- Table 2 indicates two flow rates in sccm. But what are the two gases that are flowing/being mixed?
- Figure 7 is missing label for X-axis.
Comments on the Quality of English Language
Some remarks included in the Comments and Suggestions for Authors.
Author Response
REVIEWER 3
Comments and Suggestions for Authors
The research topic may be relevant, but overall, the manuscript isn't well readable. Some substantive data is missing, as specified below. The novelty and motivation are almost absent. There are also some repetitive parts, which can be phrased more concisely.
The authors would like to thank the reviewer for the comments and suggestions to improve the paper. In the following, a point-by-point response to the queries is reported.
Motivation and novelty:
The scope of this work is the application of well-established spectroscopic techniques (Fourier transform infrared, Non-dispersive infrared and Cavity ring-down) to the analysis of the purity of parent pure gases, used in the preparation of primary certified reference gas mixtures (CRMs) by gravimetry. The preliminary results obtained in our laboratory support the great importance of this investigation for the quality, accuracy and long-term stability of the gaseous CRMs. The proposed metrological approaches represents a solid basis both for the qualification of the CRMs of CO2 at atmospheric amount fraction, and for the characterization of gas sensors for atmospheric monitoring. The final goal is indeed the application of the primary CRMs, having the highest metrological qualities, to the characterisation and calibration of gas sensors, e.g. for carbon dioxide. This motivation has been included in section 1.
- I'm a bit confused, which gases are to be considered as the main gases (i.e., intentional constituents of a prepared gas mixture) and which ones are to be considered as impurities (and why). Perhaps these two aspects are even used interchangably in the manuscript.
The gases to be considered as main gases and impurities depend on the nature and composition of the final gas mixture. In the presented cases, the certified component of the mixture is CO2 at atmospheric amount fraction (around 400-800 µmol mol-1), and the matrix constituents are Ar, O2 and N2. Indeed, CO2 is considered also as an impurity in those matrix components, as its presence could affect the final amount fraction of the gas mixture. A clarification to this point was moved in the introduction of the paper, to improve the understanding for the reader.
- I may be mistaken, but high-purity gases don't seem relevant for carbon capture and storage research (second paragraph of introduction). Maybe it would be relevant to mention some other typical applications of these traceable gas mixtures from the outset.
The references to the carbon capture and storage were removed from the text and substituted with more relevant references.
- The novelty of the work remains vague. CO2 and water vapor are rather common substances for spectroscopic investigation. Are there any improvements of accuracy or sensitivity compared to other studies?
We acknowledge that CO2 and H2O are common substances for spectroscopic investigation. Indeed, the scope of the presentend work is to investigate the application of some spectroscopic techniques (FTIR, NDIR, CRDS), in use in our laboratory, for the metrological application specifically related to impurity analysis. The basic idea is to verify the requirements of the pure gases applied in the gravimetric process for gaseous CRMs preparation, to assure quality, accuracy and long-term stability of the gaseous CRMs. This point is also of utmost importance for the participation in International Key Comparisons, to support the Calibration and Measurement Capabilities of our laboratory.
At this stage, the enhancement of accuracy and sensitivity beyond the state-of-the-art is outside the scope of the present work.
- Regarding the terminology, "amount fraction" sounds a bit clumsy to me. Isn't it the same as "molar fraction"? The term "ambient amount fraction" requires clarification as well.
- We agree that the term amount fraction is a synonym of mole fraction, and refers to the “amount of a constituent divided by the total amount of all constituents in the mixture. It is also called mole fraction”. IUPAC Green Book, (2nd ed., p. 41, https://doi.org/10.1351/goldbook.A00296). In addition, in the “SI Brochure: The International System of Units (SI)” (9th ed, 2019), the following definition is given: In the name “amount of substance”, the word “substance” will typically be replaced by words to specify the substance concerned in any particular application […] Although the word “amount” has a more general dictionary definition, the abbreviation of the full name “amount of substance” to “amount” may be used for brevity. Finally, as reported in T.J. Quinn (1998), Metrologia, 35, 807: “In chemistry, the composition of a mixture is frequently expressed by giving the value of the mass fraction or the mole fraction (which would be better called the amount fraction).”
- The term “ambient amount fraction” was substituted in the text with “atmospheric amount fraction”, and the amount fraction range was specified.
- Similarly, instead of "µmol mol-1" I would prefer "ppm" (parts per million), but the usage may depend on the research area.
In the metrology community, the use of µmol mol-1, instead of ppm, is more widespread and specific. From T.J. Quinn (1998), Metrologia, 35, 807: The terms percent (symbol %), parts per million (symbol ppm), […] are frequently used. These symbols, or pseudo units, are not part of the SI. In the physical sciences, percent and parts per million are used in this way to describe fractional parts of quantities, or fractional changes in quantities [...] The expressions percent and parts per million do not specify the quantity involved.
- I suppose NDIR and CRDS techniques involve specific wavelengths/spectral bands, where they operate (depending on the target molecule). I didn't find any information on that. Moreover, FTIR presumably involves spectral measurements, but I don't see any infrared spectra or related information. Some data could be included as supplementary material.
The data were included as supplementary material.
- Table 2 indicates two flow rates in sccm. But what are the two gases that are flowing/being mixed?
The missing information was added in the table.
- Figure 7 is missing label for X-axis.
The labels on the x-axis were added in figure 7.
Comments on the Quality of English Language
Some remarks included in the Comments and Suggestions for Authors.
The English language has been revised to improve the clarity of the text.
Reviewer 4 Report
Comments and Suggestions for Authors
Determining the purity of gases is a very important task. In this work, the mole fraction of CO2 in pure nitrogen was measured using FTIR, the mole fraction of CO2 in pure argon was measured using NDIR, and the mole fraction of H2O in several samples of synthetic and scrubbed air was measured using CRDS. All measurement results showed compliance with the data of the manufacturers of these gases. However, it is very difficult to understand what new the authors wanted to show. The fact that optical spectroscopy can be used to determine low concentrations of CO2 and H2O is well known. No new methods or details of improvements to existing techniques are presented. The paper is a description of a few routine measurements, with no evidence of scientific research or development. The introduction does not provide an overview of the possibilities and examples of application of the used spectroscopic measurement methods. The presented conclusion about the measured fraction of CO2 of 0.06 μmol mol-1 is not reliable, since the FTIR spectrometer was not calibrated for this measurement range. It is unclear why FTIR and NDIR were not used together. A study of the simultaneous use of all spectrometers (FTIR, NDIR and CRDS) for the same tasks could probably be of interest. Unfortunately, I cannot recommend this manuscript for publication in its current form.
Author Response
REVIEWER 4
Comments and Suggestions for Authors
Determining the purity of gases is a very important task. In this work, the mole fraction of CO2 in pure nitrogen was measured using FTIR, the mole fraction of CO2 in pure argon was measured using NDIR, and the mole fraction of H2O in several samples of synthetic and scrubbed air was measured using CRDS. All measurement results showed compliance with the data of the manufacturers of these gases. However, it is very difficult to understand what new the authors wanted to show. The fact that optical spectroscopy can be used to determine low concentrations of CO2 and H2O is well known. No new methods or details of improvements to existing techniques are presented. The paper is a description of a few routine measurements, with no evidence of scientific research or development. The introduction does not provide an overview of the possibilities and examples of application of the used spectroscopic measurement methods. The presented conclusion about the measured fraction of CO2 of 0.06 μmol mol-1 is not reliable, since the FTIR spectrometer was not calibrated for this measurement range. It is unclear why FTIR and NDIR were not used together. A study of the simultaneous use of all spectrometers (FTIR, NDIR and CRDS) for the same tasks could probably be of interest. Unfortunately, I cannot recommend this manuscript for publication in its current form.
The authors would like to thank the reviewer for the comments on the paper.
The scope of this work is the application of well-established spectroscopic techniques (Fourier transform infrared, Non-dispersive infrared and Cavity ring-down) to the analysis of the purity of parent pure gases, used in the preparation of primary certified reference gas mixtures (CRMs) by gravimetry. The preliminary results obtained in our laboratory support the great importance of this investigation for the quality, accuracy and long-term stability of the gaseous CRMs. The proposed metrological approaches represents a solid basis both for the qualification of the CRMs of CO2 at atmospheric amount fraction, and for the characterization of gas sensors for atmospheric monitoring. The final goal is indeed the application of the primary CRMs, having the highest metrological qualities, to the characterisation and calibration of gas sensors, e.g. for carbon dioxide. We acknowledge that CO2 and H2O are common substances for spectroscopic investigation. Indeed, the scope of the presented work is to investigate their application in the specific field of gas impurity analysis, propaedeutic to reference gas mixture realisation. This point is also of utmost importance for the participation in International Key Comparisons, to support the Calibration and Measurement Capabilities of our laboratory. The enhancement of accuracy and sensitivity beyond the state-of-the-art is outside the scope of the present work. This explanation has been included in section 1.
The presented conclusion about the measured fraction of CO2 of 0.06 μmol mol-1 is not reliable, since the FTIR spectrometer was not calibrated for this measurement range.
We acknowledge that the FTIR spectrometer was not calibrated in the very low (close to zero) measurement range. In order to estimate the impurities in the ultrapure N2, the authors decided to apply the standard addition concept*, widespread in chemical analyses, by spiking known CO2 amount fractions in the pure gas under test. This method, as mentioned in section 2.1, has several advantages, in particular the reduction of matrix effects and the possibility to carry out measurements away from the detection limit.
*Andrade, J.M.; Terán-Baamonde, J., Soto-Ferreiro, R. M.; Carlosena, A.; Interpolation in the standard additions method, Anal. Chim. Acta 780 (2013) 13– 19, http://dx.doi.org/10.1016/j.aca.2013.04.015
It is unclear why FTIR and NDIR were not used together. A study of the simultaneous use of all spectrometers (FTIR, NDIR and CRDS) for the same tasks could probably be of interest.
Many thanks for the interesting suggestion. In this work, we presented some preliminary results obtained from the application of the three measurement techniques, but we will surely try to use a combinated approach to study more deeply the impurity in pure gases.
Round 2
Reviewer 3 Report
Comments and Suggestions for Authors
- I'm still struggling to recognize the novelty/originality of the work. If the authors just introduced some well-established spectroscopic techniques for assessing the purity of the gases in their lab, it wouldn't justify writing a scientific paper. In principle, there could be some novel aspects in 1) used instrumentation; 2) used methodology/data analysis or 3) characterization results of the gases. The authors should comment on that and amend the text correspondingly.
- The captions of Table 4 and Table 5 do not specify the method that the coefficients a and b correspond to.
- Are the X-axis labels of Figure 7 some arbitrary identifiers of the cylinders or actual product names of the gas mixtures?
- "support the great importance of this investigation for the quality"
Did you mean "confirm the suitability of the methods for assessing the quality"
- There is a misplaced closing parenthesis in abstract, section 2.2 and section 3.1 (just before μmol·mol-1).
- The sentence starting with "Although the topic addressed in the manuscript is not directly related to sensors ..." is present twice in the manuscript.
Comments on the Quality of English Language
See Comments and Suggestions for Authors
Author Response
Reviewer 3
- I'm still struggling to recognize the novelty/originality of the work. If the authors just introduced some well-established spectroscopic techniques for assessing the purity of the gases in their lab, it wouldn't justify writing a scientific paper. In principle, there could be some novel aspects in
1) used instrumentation;
2) used methodology/data analysis or
3) characterization results of the gases.
The authors should comment on that and amend the text correspondingly.
Additional comments on originality of the paper were added in section 4, concerning points 2) and 3). Novel aspects in the present work are related to the methodology used for data analysis, in particular the comparison among the LPU approach and the MCM, applied to the uncertainty evaluation of the impurities of CO2 (χCO2) in the pure N2 matrix gas. In particular, the application of MCM for uncertainty evaluation in analytical chemistry applications is not yet commonly implemented, though constituting a valid alternative to the LPU approach, in particular in case of a large relative uncertainty associated with the measurement result. Unreliable expanded uncertainty intervals should be avoided and substituted with feasible coverage intervals as those provided by the MCM.
Concerning the characterization results of the gases, the applied methods allowed to confirm the high quality of the pure gases used, in particular in terms of CO2 impurities, and the compliance with the manufacturers’ specifications. This is of paramount importance for the quality and stability of the prepared CRMs. This point is also fundamental for the participation in International Key Comparisons, to support Calibration and Measurement Capabilities listed in the BIPM database (https://www.bipm.org/kcdb/cmc/quick-search), as it is not convenient to rely only on the information provided by the manufacturers.
- The captions of Table 4 and Table 5 do not specify the method that the coefficients a and b correspond to.
The captions were amended accordingly.
- Are the X-axis labels of Figure 7 some arbitrary identifiers of the cylinders or actual product names of the gas mixtures?
The X-axis labels in fig. 7 are the names assigned to the analysed gas mixtures. Those mixtures were realised in the framework of two European projects, namely the EMPIR projects 16ENV06 “SIRS” and 19ENV05 “STELLAR”, and the names were used to label the gas mixtures. This explanation was added in section 3.3.
- "support the great importance of this investigation for the quality"
Did you mean "confirm the suitability of the methods for assessing the quality"
The sentence was modified accordingly in section 1.
- There is a misplaced closing parenthesis in abstract, section 2.2 and section 3.1 (just before μmol·mol-1).
The closing parentheses were corrected.
- The sentence starting with "Although the topic addressed in the manuscript is not directly related to sensors ..." is present twice in the manuscript.
The repetition of the sentence was removed at the end of the text.

Reviewer 4 Report
Comments and Suggestions for Authors
I have not found any significant changes in the revised version of the manuscript. According to the author's response, the novelty of this work is in demonstrating the application of well-known spectroscopic methods for the analysis of the purity of parent pure gases, used in the preparation of primary certified reference gas mixtures (CRMs) by gravimetry. In my opinion, this is not enough for publication in an international scientific journal. My recommendations for improving scientific significance were not taken into account. The review of the applications of used spectroscopic measurement methods for similar tasks was not performed, data on the simultaneous use of all spectrometers (FTIR, NDIR and CRDS) were not presented, the number of references remained less than 20. I am sorry, but I cannot recommend this work for publication.
Author Response
I have not found any significant changes in the revised version of the manuscript. According to the author's response, the novelty of this work is in demonstrating the application of well-known spectroscopic methods for the analysis of the purity of parent pure gases, used in the preparation of primary certified reference gas mixtures (CRMs) by gravimetry. In my opinion, this is not enough for publication in an international scientific journal. My recommendations for improving scientific significance were not taken into account. The review of the applications of used spectroscopic measurement methods for similar tasks was not performed, data on the simultaneous use of all spectrometers (FTIR, NDIR and CRDS) were not presented, the number of references remained less than 20. I am sorry, but I cannot recommend this work for publication.
- A review of applications of spectroscopic measurement methods was added in the introduction, as suggested.
- Additional comments on originality of the paper were added in section 4. Novel aspects in the present work are related to the methodology used for data analysis, in particular the comparison among the LPU approach and the MCM, applied to the uncertainty evaluation of the impurities of CO2 (χCO2) in the pure N2 matrix gas. In particular, the application of MCM for uncertainty evaluation in analytical chemistry applications is not yet commonly implemented, though constituting a valid alternative to the LPU approach, in particular in case of a large relative uncertainty associated with the measurement result. Unreliable expanded uncertainty intervals should be avoided and substituted with feasible coverage intervals as those provided by the MCM.
Concerning the characterization results of the gases, the applied methods allowed to confirm the high quality of the pure gases used, in particular in terms of CO2 impurities, and the compliance with the manufacturers’ specifications. This is of paramount importance for the quality and stability of the prepared CRMs. This point is also fundamental for the participation in International Key Comparisons, to support Calibration and Measurement Capabilities listed in the BIPM database (https://www.bipm.org/kcdb/cmc/quick-search), as it is not convenient to rely only on the information provided by the manufacturers.
- The authors are sorry that the current version of the paper does not meet the expectations of the reviewer. Unfortunately, due to the limited time for the revision of the paper, it is not possible for the authors to carry out the suggested additional measurements to collect data on the simultaneous use of all spectrometers (FTIR, NDIR and CRDS) and to incorporate them in the present paper. We will consider the suggested measurements for future developments of the experimental work.
